# The Anatomy of Health-Supportive Neighborhoods: A Multilevel Analysis of Built Environment, Perceived Disorder, Social Interaction and Mental Health in Beijing

**DOI:** 10.3390/ijerph17010013

**Published:** 2019-12-18

**Authors:** Yinhua Tao, Jie Yang, Yanwei Chai

**Affiliations:** College of Urban and Environmental Sciences, Peking University, Beijing 100871, China; yh.tao@hotmail.com (Y.T.); syexyj@163.com (J.Y.)

**Keywords:** built environment, neighborhood disorder, social interaction, mental health, Beijing

## Abstract

Mental health is an exceedingly prevalent concern for the urban population. Mounting evidence has confirmed the plausibility of high incidences of mental disorders in socioeconomically disadvantaged neighborhoods. However, the association between the neighborhood built environment and individual mental health is understudied and far from conclusive, especially in developing countries such as China. The underlying mechanism requires in-depth analysis combining potential intermediates such as perceived environmental disorder and supportive social relationships. Using a health survey conducted in Beijing in 2017, this study investigates for the first time a socio-environmental pathway through which perceived disorder and social interaction account for the relationship between the built environment and mental health under the very notion of the neighborhood effect. The results from multilevel structural equation models indicate that individual mental health is influenced by the neighborhood-scale built environment through three pathways, independent of neighborhood socioeconomic disadvantages: (1) proximity to parks is the sole indicator directly linked to mental health; (2) population density, road connectivity and proximity to parks are indirectly associated with mental health through interactions with neighbors; and (3) population density, road connectivity and facility diversity are partially associated with perceived neighborhood disorder, which is indirectly correlated with mental health through interactions with neighbors. This study is a preliminary attempt to disentangle the complex relationships among the neighborhood environment, social interaction and mental health in the context of developing megacities. The relevant findings provide an important reference for urban planners and administrators regarding how to build health-supportive neighborhoods and healthy cities.

## 1. Introduction

Mental health is an increasingly pressing issue in developing megacities. It was estimated that 173 million patients suffer from diagnosable psychiatric disorders in China, accounting for nearly one-quarter of the total disease burden [1]. Pervasive mental health problems not only impair the public’s quality of life but also impose great financial burdens on city governments. Given the multilevel structure of individual health determinants [2], the renewal and renovation of the neighborhood environment is widely accepted as a realistic strategy to improve mental health beyond unchangeable individual characteristics. Existing studies have found that residential social contexts are associated with mental health. Socioeconomically deprived neighborhoods, such as low-income and racial minority communities, are likely to have a higher prevalence of depression, anxiety and other psychological symptoms [3,4,5]. In contrast, it is under addressed and inconclusive whether the neighborhood-scale built environment is a relevant geographic context for explaining individual mental health. In this study, two distinctive but correlated intermediates are proposed to dissect the relationship between the implicit built environment and mental health in light of the neighborhood effect. First, daily behavioral interactions with neighbors may provide individuals with supportive social ties to cope with mental disorders [6]. In Chinese megacities, however, traditional cohesive neighborhoods are gradually being replaced by gated communities with limited communal space and pedestrian-oriented designs. The scarcity of space has led to the sacrifice of neighborly relations for the sake of privacy and anonymity, which may generate a sense of loneliness and helplessness [7]. Second, the perception of environmental stressors and disorders is connected with, albeit not entirely dependent on, physical settings [8]. The neighborhood built environment may influence residents’ initial feelings of safety and disorder, and this emotional response is likely to improve or harm residents’ social relationships and mental states correspondingly. Thus, this study intends to clarify the complex relationships among the built environment, perceived disorder, social interaction and mental health in a developing megacity, Beijing, China. Drawing upon a health survey dataset collected in 2017, multilevel structural equation models are employed to examine the direct effect of the neighborhood built environment on individual mental health, as well as the mediating role of perceived disorder and social interaction.

## 2. Literature Review

### 2.1. Neighborhood Environment and Mental Health

According to the socioecological model, mental health is shaped and reshaped by a multilevel structure of factors, including individual compositional effects and environmental contextual effects [2]. Environmental research—which includes research on the built environment and socioeconomic conditions—often treats the neighborhood as the geographic context to detect how residential locations and exposure to the surrounding environment trigger or relieve related psychiatric disorders, such as depression, anxiety and schizophrenia [9,10]. An influential review even claimed that the housing and neighborhood environment was becoming more relevant to mental health than exposure to individual environmental stressors [11].

A wide array of studies has documented that residing in a disadvantaged neighborhood comprised of socioeconomically deprived residents increases the risks of multiple mental health problems [12]. From the perspective of social disorganization, such deprived neighborhoods are deficient in the necessary economic resources and political power to implement social control on environmental risks, and weak social support further deteriorates residents’ vulnerability to mental illnesses [13,14]. Residents from socioeconomically disadvantaged neighborhoods are also confronted with structural barriers, such as poverty, long-term unemployment and dilapidated housing [5]. For instance, Silver, Mulvey and Swanson (2002) found that disadvantaged neighborhoods, characterized by a high proportion of female-headed and low-income households, were associated with higher rates of major depression [3]. A systematic narrative review suggested that migrants concentrated in rundown residential areas had high levels of anxiety symptoms and post-traumatic stress disorders [4].

Regarding the health effect of neighborhood built environments, an increasing amount of empirical research has focused on health-related behaviors (e.g., physical activity, healthy dieting, smoking and alcoholism) and their physical health outcomes. The findings are relatively consistent in showing that a return to traditional compact development is conducive to curbing car use, enhancing exercise intensity and developing healthy lifestyles, thereby reducing the morbidity of obesity, cardiovascular diseases and other noncommunicable diseases [15,16,17]. The key characteristics of the built environment underpinning compact development include high residential density, mixed land use, grid-like and small-block street networks, and easy access to diverse destinations [18].

The connection between the built environment and mental health from the neighborhood perspective is a subject of emerging interest. A growing body of studies to date has examined the mental health effects of certain built environment indicators, but their findings are still limited and far from conclusive. For example, some evidence suggested that high population density was a surrogate for urban vitality and sufficient opportunities to help relieve psychotic disorders [19]. Neighborhoods with well-connected streets and diverse land use enabled residents to have more route or destination choices for leisure activities and to build close community ties, which proved beneficial in reducing the risk of depression [20,21]. In contrast, another study contended that overpopulation was predictive of a crowding effect, which caused greater physiological stress [22]. Dense road networks and high accessibility to facilities were shown to be much more important for physical fitness, while their psychological benefits were negligible [23]. Notably, it was acknowledged that exposure to natural space might be associated with positive psychological status, both directly (e.g., recovery from cognitive fatigue and stress) and indirectly (through encouraging physical activity and intensifying social network) [24,25], but a recent review indicated that the results from prior studies were constantly contradictory and mostly weak in effect [26].

The above inconsistency can be ascribed to the complex interactions between the neighborhood environment and mental health. The neighborhood environment involves a subset of built environment attributes and socioeconomic conditions. Neither of them influences mental health independently; rather, they are intertwined with each other. For instance, a poorly maintained built environment may be a reflection of potential socioeconomic disadvantages, which synthetically weakens the capacity for defense against health risks. To date, however, few studies have included these multidimensional built and social environment characteristics within a holistic framework. In addition, the association between the built environment and mental health may be more implicit than the association between the built environment and physical health. Thus, indirect pathways are hypothesized to be more plausible, such as through perceptions of environmental stressors and disorders, supportive social relationships, and neighborhood attachment and belongingness [10,27]. Additionally, many studies are limited by poor methodological and study designs, such as those that combine objective measures and subjective evaluations of the built environment and ignore the multilevel structure of contributors to mental health.

### 2.2. Built Environment, Perceived Disorder and Mental Health

A key issue in analyzing the connection between the environment and health is how environmental attributes should be delineated and measured at the neighborhood scale. Most studies have relied on two methods of measurement: objective measures of physical settings and subjective evaluations of living environments [8]. It is often the case that objective markers do not align with residents’ self-reported exposure, and the latter is found to be more correlated with self-reported psychological problems, such as overstress, depressive symptoms and self-rated mental disorder [28,29]. However, the stronger health effect of environmental appraisals may be spurious. One reason is that same-source bias may occur when both built environment and health outcomes stem from the reports of the same individual [30]. For example, people with mental illnesses are inclined to rate their surroundings to be health-threatening, and vice versa. For this reason, the use of perceptions of the environment alone or the combination of objective and subjective measures of the built environment may distort findings regarding whether objective contextual constructs or subjective predispositional appraisals are valid indicators of mental health.

We propose that perceived neighborhood disorder may mediate the association between the objective built environment and mental health. Perceived disorder refers to “the perceived lack of order and a weak social control with observable signs and visible cues” [31]. In disordered neighborhoods, residents tended to feel unsafe and noisy under the attack of the chaotic environment and deviant behaviors [32,33]. Several studies also showed that residents living in disordered environments were reluctant to communicate with neighbors and faced higher levels of incivility [34,35]. The rationale for the health effect of perceived disorder has been expounded through three distinct pathways [36,37,38]. First, the psychosocial pathway is the mechanism through which senses of unsafety and disorder increase chronic biological stress, which erodes mental health. The second explanation concerns the adverse effects of perceived disorder on emotional orientations and health-related behaviors, which are indicative of psychological distress. For instance, the perceived scarcity of public space deterred neighbors from going outside to communicate and resulted in higher levels of mistrust and fear [39]. Third, external social groups are reluctant to come in contact with people from disordered neighborhoods, which may intensify social isolation and discourage economic investment in building health-supportive neighborhoods.

Some studies have examined the extent to which neighborhood built environments are associated with perceived disorder, but few have extended their examinations to health outcomes, especially psychological reactions. For instance, a positive aspect of densely populated and well-connected streets is the creation of a pedestrian-oriented environment with surveillant “eyes on the streets” [40], but this positive aspect is coupled with latent health risks when streets are arranged in a chaotic and noisy way [41]. Mass et al. (2009) suggested that abundant green space in the living environment could enhance residents’ feelings of social safety, but the opposite effect was observed in highly dense areas [42]. Brown and Lombard (2014) discovered that a variety of retail and public facilities made neighborhoods more interesting and recognizable [43]. Residents accordingly found more opportunities to chat with friends nearby and improve their sense of belonging and well-being. As a result, it seems plausible that the built environment is not only directly related to psychological status, as suggested, but also influences residents’ perceptions of environmental disorder and, in turn, explains mental health outcomes to some degree.

### 2.3. Built Environment, Social Interaction and Mental Health

Social interaction is another aspect that links neighborhood built environments with mental health. A great deal of research has shown that social interaction in living environments, especially casual interactions between neighbors, generates considerable psychological benefits [44,45,46]. From a cognitive perspective, informal social contacts help residents build solid social ties with community members. This conscious or unconscious social support leads to the creation of a friendly neighborhood atmosphere with high levels of trust and reciprocity, protecting residents from pathological mental states such as depression and anxiety [43]. From a structural perspective, frequent behavioral interactions are a prerequisite for strong social cohesion and collective efficacy—both of which relate to translating individual social ties into collective common goals to combat health risks [47]. A cohesive neighborhood also better disseminates health-related information and engages in health-promotive behaviors [48]. A population-based cohort study found that less socially cohesive neighborhoods were associated with a high prevalence of self-reported depressive symptoms [27]. Furthermore, Erdem et al. (2016) discovered that living in cohesive neighborhoods alleviated psychological distress among economically deprived groups [49].

Although the majority of evidence suggests that the built environment provides opportunities for or constraints to social interaction, it is unclear whether and to what extent social interaction accounts for the environment–health relationship. On the one hand, a walkable environment encourages neighbors to have more informal contacts. Neighborhoods with dense populations, short blocks and mixed-use zoning may encourage people to participate in group exercise and make acquaintances with each other [50,51]. Additionally, diverse destinations around residences, such as parks, squares, and cultural and commercial facilities, are important locales for the formation of social ties [5,7]. On the other hand, a car-supportive environment discourages neighbors from establishing social relations and leads to social isolation. Cabrera and Najarian (2015) discovered that adults living near main roads were disturbed by the heavy traffic volume and were less likely to know their neighbors [44]. The presence of public parking lots is also related to declining perceptions of neighborhood friendliness [52], whereas proximity to public transit is conducive to casual connections with neighbors [53]. Moreover, an interdisciplinary review argued that while physically deteriorated and pedestrian-unfriendly neighborhoods incur excessive environmental stressors that are detrimental to mental health, residents living in such neighborhoods are likely to increase, rather than restrict, their development of social relationships as a coping strategy against insufficient environmental resources [54]. In other words, social interaction is still an ambiguous pathway linking the built environment and mental health, which requires further examination.

To further elaborate, perceived disorder and social interaction are not independent constructs in exploring the association between the built environment and mental health. There is scant evidence, however, whether perceptions of neighborhood disorder erode the structure of social relationships and then trigger mental illnesses. If residents feel unsafe, noisy and disordered in their neighborhoods, they may be less likely to interact and engage in reciprocal relationships with neighbors. Consequently, weak and mistrustful social networks may be linked to passive emotional states such as tiredness, suspiciousness and helplessness. Kim (2010) provided empirical evidence that neighborhood socioeconomic disadvantages (a latent construct of the prevalence of female-headed households and poverty households) increased psychological distress through enhanced perceived disorder and restricted social relationships. Meanwhile, perceived disorder was also indirectly associated with psychological distress through social relationships. However, this finding may not have been complete and robust for two reasons. First, the health effect of the built environment was not taken into account. Second, the social relationship items were not well defined among neighbors but were among the general population.

In summary, the relationship between the built environment and mental health has been of particular interest in recent years. Prior studies have produced limited and inconclusive results because few have paid attention to the internal mechanism to disentangle the environment–health relationship. To the best of our knowledge, this is the first study investigating a socio-environmental pathway through which social interaction accounts for the effects of the neighborhood-scale built environment and perceived disorder on individual mental health in the context of developing megacities. The conceptual framework is shown in Figure 1, which features the following four research questions (RQs):RQ 1: Is the neighborhood built environment associated with mental health after controlling for neighborhood socioeconomic disadvantages?RQ 2: Does perceived neighborhood disorder explain the association between the built environment and mental health?RQ 3: Does social interaction with neighbors explain the association between the built environment and mental health?RQ 4: Based on the relationship between the objective built environment and perceived neighborhood disorder, does social interaction with neighbors further explain the association between perceived disorder and mental health?

## 3. Data and Methods

### 3.1. Data

Beijing, a developing megacity, is undergoing tremendous urbanization and industrialization, during which environmental exposure, social isolation and public health have become salient concerns for urban planners and administrators. In this study, a health survey was conducted in Beijing in 2017 to examine neighborhood environments, social interaction with neighbors, and their associations with self-reported mental health symptoms after adjusting for individual and contextual socio-demographics and residential stability.

A spatial stratified random sampling procedure was used. In the first step, 26 residential communities [1] with a wide range of spatial locations, built ages and housing conditions were randomly selected (Figure 2). These chosen communities were representative of the main types of urban neighborhoods in Beijing, including commodity housing neighborhoods, work-unit compounds (or *danwei*, the neighborhood where workplace and housing are organized as a spatial unit by governments), policy related housing neighborhoods and mixed neighborhoods. In each neighborhood, approximately 50 residents were randomly selected based on the building and house number to participate in in-person interviews. Inclusion in the sample was limited to only one household member aged 18–65 years old in each family. The survey was approved by each district government, and all participants gave written informed consent before the interview. Data confidentiality is strictly implemented in accordance with the Chinese Personal Information Protection Regulations.

In total, 1300 residents were recruited for our survey, and 1280 provided valid responses, for a good response rate of 98.5%. To avoid the effect misestimate of the short-term exposure on long-term health status [24], only respondents residing in the surveyed neighborhoods for more than 1 year were included in the analysis, which left 1256 respondents whose responses were used to form our final database. Table 1 presents the key sociodemographic attributes of the respondents: 50.7% were males, 74.9% were locals, and 58.0% had full-time jobs. The age and household income distributions were well balanced, and the mean duration of residency was 13.79 years. The samples were representative of the general populations at the time of 2010 population census data.

In addition to the health survey, 2017 population statistical data acquired from the official neighborhood committees were used to measure neighborhood-level sociodemographic disadvantages, including the prevalence of elderly people, illiterate people and rural-to-urban migrants. Meanwhile, point-of-interest (POI) data [2] in the year of 2014 was derived from the Amap corporation, one of the most reliable authorities in digital maps in China, and then employed to measure multi-dimensional objective measures of neighborhood built environment, including population density, road connectivity, facility proximity and availability. We consider the temporal discrepancy between the survey data and POI data acceptable because built environment characteristics are relatively stable in the short term.

### 3.2. Measurement

Mental health was measured by the self-reported frequency of psychological distress, which incorporated 4 dimensions of subclinical symptoms: anxiety, overstress, fatigue and headache. Respondents were required to rate how often they had suffered from the abovementioned mental disorders over the past half year. Each dimension was quantified on a 4-point scale (1 = never, 2 = rarely, 3 = sometimes, and 4 = often) and was reverse scored, so that a higher score indicated a better health status. Consistent with a previous study [55], these 4 dimensions loaded well on the latent construct of mental health. To test the robustness of this latent construct, two additional analyses were conducted. First, each dimension of mental health was analyzed separately to determine whether the environment–health relationship differed by each symptom. Second, the mental health outcome was replaced by a single item from the overall mental health assessment. The findings from the sensitive analysis showed no significant differences in the results of the latent mental health variable.

Social interaction was measured by the self-reported frequency of communication with neighbors, which has been employed in a number of scholarly publications [56]. Respondents were prompted to answer the following questions: How often (1) have you visited neighbors, (2) have you and your neighbors helped each other, (3) have you had meals with neighbors, and (4) have you exercised with neighbors during the past half year. Each item was estimated on a 4-point scale (1 = never, 2 = rarely, 3 = sometimes, and 4 = often), with a higher score representing greater neighborhood social ties. These four items loaded well on the latent construct of neighborhood social interaction.

Perceived disorder was measured by a modified and abbreviated version of Ross-Mirowsky’s physical and social disorder scale [31]. Respondents were asked to rate the following three items on a 4-point scale from 1 (strongly agree) to 4 (strongly disagree): (1) My neighborhood is quiet and tidy; (2) My neighborhood is safe; and (3) There are many places and opportunities in my neighborhood to communicate with neighbors. A higher score on each item indicated a worse perception of neighborhood disorder. These three items loaded well on the latent construct of perceived neighborhood disorder.

The built environment concerning social interaction and mental health was measured at the neighborhood level, including population density, road connectivity, facility diversity, proximity to parks and availability of rail stations and parking lots. An 800 m buffer around the community centroid was used to define the boundary of each neighborhood since this area reflected the standard walkable catchment distances widely adopted by prior studies [57,58] and formulated in the Beijing Urban Master Planning (2016–2035). Descriptions of the built environment characteristics stratified by neighborhood types are displayed in Table 2. Compared with commodity housing neighborhoods, other types of neighborhoods were more overpopulated with denser road networks and better access to rail stations, while policy related housing neighborhoods lacked access to parks, parking lots and living facilities. Work-unit compounds provided the most pedestrian-oriented environments with the fewest parking lots and the greatest accessibility to diversified destinations. Notably, there was high collinearity between the availability of rail stations (1 = no) and parking lots (1 = yes); hence, a composite index, i.e., access to car-supporting facilities, was calculated as the sum of both scores (1 = having no access to parking lots and rail stations, 2 = having access to either parking lots or rail stations, and 3 = having access to both parking lots and rail stations).

Neighborhood socioeconomic disadvantages were measured by the proportion of rural-to-urban migrants, old people aged over 65 years old, and residents with high school degree and below in the neighborhood. These three indices have been used in Chinese urban studies and environmental justice research to depict the extent of disadvantaged social contexts [7,59,60,61]. As shown in Table 2, there was a lower prevalence of migrants and a higher prevalence of elderly people in work-unit compounds and mixed neighborhoods than in commodity housing and policy related housing neighborhoods. Besides, a higher proportion of residents with the education level of high school and below was found in work-unit compounds and policy related housing neighborhoods. 

The confounders were individual and household socio-demographics, including gender (1 = male, 0 = female), employment status (1 = full-time employees, 0 = others), household monthly income (a categorical variable, where 1 = below 5000 RMB, 2 = between 5000 and 15,000 RMB, and 3 = above 15000 RMB), and duration of residency (in years). Given that 66 respondents had missing values for duration of residency, full information maximum likelihood (FIML) was employed to iteratively estimate the parameters. This method has advantages in retaining the information of all observed variables and producing unbiased and valid estimate values.

### 3.3. Statistical Model

Multilevel structural equation models (SEMs) were constructed to examine the complex environment–contact–health relationship. This modeling method could not only disentangle the association between the latent constructs with measurement errors (mental health, social interaction, and perceived disorder) and the observed indicators, but also compute the direct and indirect effects of various determinants on health outcomes. Since social interaction and mental health are the result of a series of factors operating at both the individual and neighborhood levels, as suggested by the socioecological model, the multilevel modeling method was combined with structural equation modeling to simultaneously discern within-neighborhood variations (conditioned by individual characteristics) and between-neighborhood variations. Robust maximum likelihood estimators (MLRs) were used to address the non-normal and non-independent data structure. This estimator is unbiased and robust in the estimation of medium–small datasets.

After ascertaining the validity of the latent constructs, we fitted four multilevel SEMs with increasing complexity to test the four aforementioned research questions (RQs), as shown in Figure 3. The baseline model (Model 1) only included the sociodemographic confounders and the neighborhood socioeconomic disadvantage variables, in line with conventional research practices [3,4,12]. Model 2 added the neighborhood-level built environment characteristics as explanatory variables to examine the environment–health connection (RQ1). The latent construct of perceived neighborhood disorder was incorporated in Model 3 to explore its mediating role (RQ3). In the fully adjusted model (Model 4), the latent social interaction index was taken into account to determine whether it explained the pathways linking the built environment and mental health (RQ3), and perceived disorder and mental health (RQ4).

Four commonly used indices were calculated to assess the goodness of fit of the model, i.e., χ^2^ with the degrees of freedom (d.f.), Comparative Fit Index (CFI), Root Mean Square Error of Approximation (RMSEA) and Standardized Root Mean Square Residual (SRMR). The appropriate thresholds are CFI > 0.90, RMSEA < 0.05 and SRMR < 0.05. The Intra-Class Correlation Coefficient (ICC) was also provided to indicate the ratio of the total variance explained by neighborhood-level effects. ICC > 0.02 is considered acceptable. All the models were implemented in Mplus 8.3.

## 4. Results

### 4.1. Validity Test of the Latent Constructs of Perceived Disorder, Social Interaction and Mental Health

Table 3 displays the unstandardized and standardized loadings for the measurement models. All the standardized factor loadings are above 0.60, with the factor loadings and measurement errors significant at the *p* < 0.01 level. This result suggests that the underlying indicators manifest the intended latent constructs—that is, perceived neighborhood disorder, neighborhood social interaction, and mental health. The proposed constructs are reliable and valid for the following multilevel SEM analysis.

### 4.2. Association between Neighborhood Environment and Mental Health

Table 4 presents the multilevel SEM results for mental health. Consistent with previous findings, Model 1 shows that a socioeconomically disadvantaged neighborhood has a higher likelihood of residents having mental disorders, after controlling for individual and household demographics. Individuals living in a neighborhood with a higher concentration of rural-to-urban migrants and well-educated residents are more likely to have worse mental health, but the effect of the proportion of elderly people is nonsignificant. Model 2 examines whether the built environment is associated with mental health to answer research question 1. The result indicates that built environment characteristics only exert a modest effect on the mental health outcome: a better connected road network is predictive of more frequent mental disorders, while closer proximity to parks is beneficial to mental health. The standardized direct effects are −0.330 and −0.439 for road connectivity and proximity to parks, respectively. Although diverse public and commercial facilities and lack of car-supporting facilities are conducive to a positive and healthy mindset, their effects are not significant.

### 4.3. Mediation of Neighborhood Perceived Disorder and Social Interaction

Table 5 shows the multilevel SEM results for perceived disorder, social interaction and mental health. For better interpretations, Figure 4 provided the significant results of the neighborhood environment-social interactions-mental health relationships in the fully adjusted model (i.e., Model 4). Model 3 is built to test research question 2 on whether perceived neighborhood disorder serves as a mediator between the built environment and mental health. The result fails to confirm a mediating role of perceived disorder given its nonsignificant correlation with mental health (b = −0.110, St. Err. = 0.137). Another evidence is that the inclusion of perceived disorder marginally changes the health effect of the built environment compared with that in Model 2. However, the objective built environment is related to perceived neighborhood disorder for the most part. More specifically, a densely populated neighborhood with well-connected main roads but lacking diverse facilities increases the likelihood of perceived disorder of the neighborhood environment.

Research question 3 refers to whether interaction with neighbors mediates the association between the built environment and mental health. As shown in Model 4, the mental health score increases by 11.7% with each additional increase in the standard deviation of social interaction, confirming the direct health effect of frequent communication with neighbors. Furthermore, a densely populated and well-connected neighborhood with low proximity to parks significantly restrains the development of neighboring relationships, indicating that social interaction mediates the effect of the built environment on mental health. After introducing social interactions in Model 4, the direct health effect of road connectivity becomes nonsignificant, while that of proximity to parks remains significant but decreases in magnitude by 44.8% [(0.440–0.243)/0.440]. Notably, the availability of car-supporting facilities (including parking lots and transit stations) around the neighborhood decreases interpersonal relationships and psychological well-being simultaneously, albeit not significantly.

While perceived neighborhood disorder fails to mediate the association between the built environment and mental health in Model 3, it remains unclear whether social interaction explains the pathway from perceived disorder to mental health as proposed in research question 4. The results of Model 4 suggest that with each additional increase in the standard deviation of perceived disorder, the social interactions score decreases by 31.6%, validating the indirect health effect of perceived disorder [(−0.316) × 0.117 = 0.037] through interactions with neighborhoods. Overall, in the fully adjusted model, the built environment is associated with mental health in three ways: (1) proximity to parks is the sole indicator with a direct and independent effect on mental health, (2) several built environment indices (population density, road connectivity and proximity to parks) have indirect effects on mental health via social interaction, and (3) some built environment elements (population density, road connectivity and facility diversity) are associated with perceived neighborhood disorder, which has an indirect effect on mental health via social interaction.

A final note is that every model in Table 4 and Table 5 fits the data well, as all goodness-of-fit indices (χ^2^, CFI, RMSEA and SRMR) are within the acceptable boundaries. The ICCs for perceived disorder, social interaction and mental health are 1.160, 0.056 and 0.079, respectively, indicating that their variations for individuals from the same neighborhood are relatively clustered. The use of multilevel SEMs is appropriate. 

## 5. Discussion and Conclusions

### 5.1. Discussion and Limitations

This study explores the association between the neighborhood built environment and individual mental health, along with the mediating effects of perceived neighborhood disorder and social interaction. The four research questions mentioned above are answered to varying degrees. First, certain built environment attributes at the residence level, including proximity to parks and road connectivity, are correlated with mental health outcomes, independent of neighborhood socioeconomic disadvantages. In addition, perceived disorder is not a valid mediator of the relationship between the built environment and mental health, while social interaction with neighbors significantly mediates the relationship between mental health and three indicators of the built environment (population density, road connectivity and proximity to parks). Finally, after the link between the objective built environment and perceived neighborhood disorder is validated, perceived disorder is further correlated with mental health via social interaction with neighbors. The above findings provide a new perspective for explaining the mechanisms by which neighborhood-scale environmental characteristics affect individual mental health among the general population in developing megacities.

Consistent with previous studies [28,29], the neighborhood built environment is moderately associated with individual mental health. This finding confirms that, unlike physical health, mental health is more implicitly influenced by the neighborhood environment [10]. Its inherent mechanisms are complicated, and so relevant studies need to test and clarify possible intermediates, such as perceptions of environmental stressors, neighborhood attachment and belonging, social ties and capital. In this study, we find that after controlling neighborhood perceptions and social interaction, proximity to parks is the sole indicator exerting a direct and independent effect on mental health, which indicates the immense psychological benefits of exposure to large-scale green space around neighborhoods [24]. Apart from encouraging frequent social contact, green space creates opportunities for people to be close to the natural environment, which is a scarce resource in a brick-and-mortal megacity. Urbanites travel through high-rise buildings and crowded urban environments with burdensome errands and tight timetables on a daily basis. Park visits during free time or merely gazing at greenspace at a distance provide a sense of peace and tranquility, helping urbanites conquer stress and attention fatigue [62]. Notably, there may be other indirect pathways linking access to park and mental health, such as the well-documented role of physical activity [24], but examining other indirect pathways is beyond the scope of our study.

A novel finding is that certain built environment attributes induce residents’ perceptions of neighborhood disorder. This result suggests that past research mixing objective measures and subjective evaluations as a unidimensional subset of built environment indices may have been problematic [8]. In particular, the significant connection between the perceived environment and mental health may mask the actual health effect of the objective built environment. Our study shows that a densely populated and well-connected neighborhood is correlated with perceived disorder, which suggests that a crowding effect would occur if the density, whether the population or traffic density, were to exceed a certain threshold [43]. A dense stream of unfamiliar people and auto traffic engenders an unsafe and noisy atmosphere. To some extent, the broken windows theory applies, as a deprived built environment transmits a message about the lack of order and social control, thereby generating a sense of mistrust and fear among residents [63]. By contrast, great accessibility to diverse commercial and public facilities improves the orderliness of neighborhoods. Even though those facilities are possibly the main source of neighborhood “noises” illustrated by Ma et al. (2018) [55], residents do not regard them as chaotic and noisy. The insertion of diverse facilities into neighborhoods, on the contrary, leads to frequent lingering and walking behaviors, enhances the recognizability of neighborhoods and guarantees more “eyes on the street,” as advocated by Jacobs’s important work (1961) [64]. Unfortunately, the results do not demonstrate that perceived neighborhood disorder mediates the relationship between the built environment and mental health, which requires future studies to provide a rational explanation.

Informal social interaction with neighbors is found to be a key mediating pathway linking the neighborhood built environment to individual mental health. More specifically, high population density and road connectivity are negatively associated with mental health through less frequent social interaction. Especially for road connectivity, the mental health effect is fully mediated by social interaction with neighbors. This finding partially contradicts previous conclusions regarding Western cities, where low-density and auto-oriented sprawling patterns are the common case. In the context of developing megacities, rapid urban expansion takes the form of high-density compact development under the dual forces of the market and government [65]. Notably, the mean population density and road connectivity values are above 80 people and 100 intersections for each neighborhood unit (800 m buffer around the neighborhood centroid), respectively (Table 2). The crowding effect holds true when residents are exposed to a large number of unfamiliar social interactions, and these unwanted interactions may frustrate and exhaust them psychologically. In addition, we note that road connectivity in our study is measured by the density of city main roads, and this limited measure is more related to car travel than walking behavior. Consistent with previous findings [16,66], an auto-oriented environment discourages people from daily communication with neighbors and consequently constitutes a threat to mental health. Future research should include a detailed classification of urban road networks, especially examining the mental health effects of sidewalks and pedestrian crossings. Finally, high proximity to parks is shown to promote residents’ psychological gains via casual interactions with neighbors. Although recent studies show that neighborhood public space and semipublic space may be more favorable sites for neighbors’ everyday contacts [67], this study argues that parks should still serve as a locus of social interaction due to their remarkable psychological healing and stress buffering effects.

Although our study does not indicate that perceived neighborhood disorder poses a direct threat to mental health, it shows that neighborhood disorder does have an indirect health effect mediated by social interaction. Perceived safety and noise issues are likely to create barriers to residents’ daily social interactions. If residents feel threat or disorder from neighbors and the surrounding environment, they may be reluctant to communicate and develop social ties with others. A lack of social support and mutual trust in the living environment results in psychological distress and fatigue. In addition, perception of available places for interactions, an item for the perceived disorder score, is a better proxy for actual communication opportunities than the geographic accessibility of facilities. If residents feel comfortable and satisfied in the neighborhood contact space, they are more likely to form supportive relationships and have better psychological states. Future studies would benefit from examining the reason for the mismatch among the availability of, practical use of and satisfaction with contact space. Some possible explanations include economic access and entry, microenvironmental design and layout, and information and knowledge of the surroundings.

Several limitations should be borne in mind during the interpretation of our research findings. First, the cross-sectional design limits the ability for causal inference regarding the relationship between the built environment and mental health. The reverse direction is possible, i.e., that mentally unhealthy individuals are concentrated in disadvantaged neighborhoods where the built environment is likely to be perceived as disorderly and unfavorable to social interactions. In this study, individual and household socio-demographics and duration of residency were controlled to minimize self-selection effects, but longitudinal datasets and experimental designs are still necessary to shed light on the causality in the relationship between the environment and health. Second, this study selects Beijing as the case of developed megacities in China, and so the environment–health relationship is still unclear in rural areas, and small and medium-sized cities. The extent to which the findings can be generalized to other cities requires further cross-validation. Third, social interaction in this study represents informal “bonding ties”—ties that connect individuals from similar social and geographic spheres—constructed between neighbors. However, an ego-based network analysis by Cabrera and Najarian (2015) suggested that intergroup contacts beyond the limitation of neighborhood boundaries—called bridging ties—were more conducive to social cohesion and the accumulation of social capital [44]. Future studies should develop a more comprehensive assessment of social interactions that differentiates bonding ties and bridging ties, as well as compare their associations with environment and health.

### 5.2. Conclusions and Implications

The relationship between the built environment and mental health is a burgeoning but controversial concern in the field of health geography and urban planning. Employing multilevel structural equation models, this study provides empirical evidence by examining the mediating role of perceived disorder and social interaction. Overall, three pathways linking the built environment and mental health are clarified: (1) the built environment is weakly and directly associated with mental health; (2) the built environment is indirectly associated with mental health through social interactions; and (3) the built environment is partially associated with perceptions of neighborhood disorder—the latter of which is further associated with mental health through social interaction.

The current findings have policy implications for urban planning and neighborhood design. First, residential planning needs to change from a focus on physical space to a focus on interpersonal contact space. Communal space for communication should be well reserved in neighborhoods to encourage residents to engage in casual social interactions. It is recommended that the distribution of communal space should avoid population and traffic overcrowding, such as through the coordinated development of scattered space to accommodate more people and avoiding city main roads. Second, great importance should be attached to the design of urban greenness and public space, as well as circumventing auto-oriented neighborhood environment. Especially, planning of sites for parks should combine the principle of concentration and dispersion. On the one hand, gated neighborhoods should be eliminated to improve the sharing and openness of green space. On the other hand, in the context of insufficient provision of neighborhood-scale green space in developing megacities, it is more feasible to advocate for scattered pocket parks to increase overall accessibility for each neighborhood. Third, urban planners should look beyond compliance with planning standards simply to achieve the proximity and accessibility of physical settings. It is residents’ experiences and satisfaction that is important for public health and well-being, and so their subjective evaluations of the neighborhood environment should be taken as essential indicators in the planning process.

The relationships among the built environment, perceived disorder, social interaction and mental health make for a complex anatomy. This study is a preliminary attempt to understand why and how the neighborhood built environment is a contextual contributor in reshaping individual mental health from the perspective of interaction with neighbors and perceptions of neighborhood disorder. It is our sincere hope that close collaboration be developed among urban planners, public health professionals and general urban populations to create health-supportive neighborhoods and healthy cities.

## Figures and Tables

**Figure 1 ijerph-17-00013-f001:**
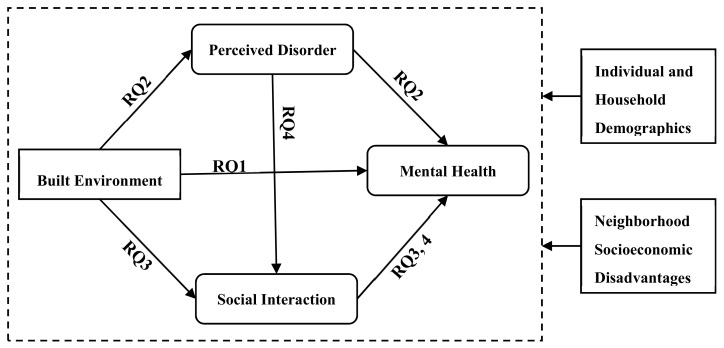
Conceptual framework. RQ = research question.

**Figure 2 ijerph-17-00013-f002:**
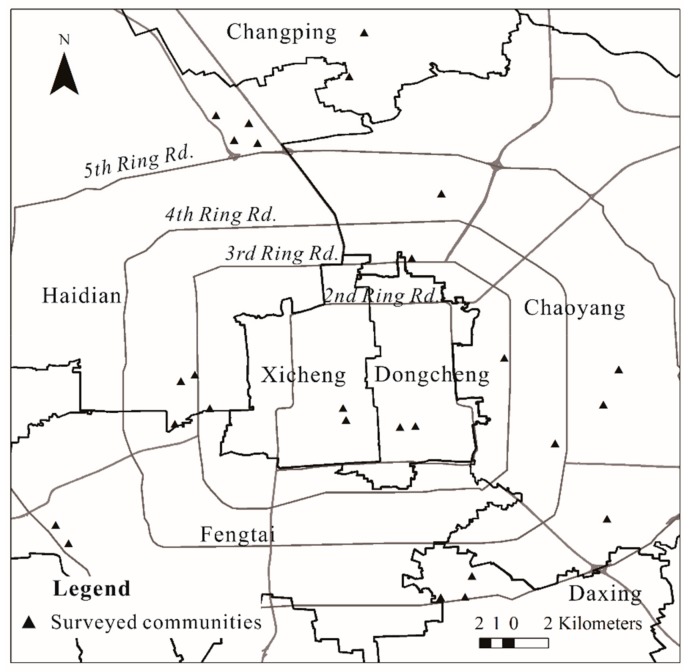
Distribution of surveyed communities.

**Figure 3 ijerph-17-00013-f003:**
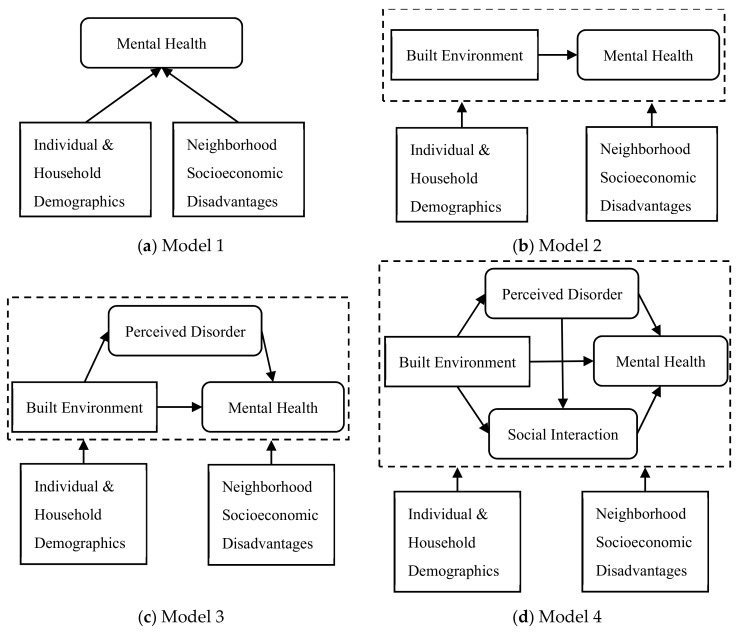
Specifications of multilevel structural equation model (SEM) structures.

**Figure 4 ijerph-17-00013-f004:**
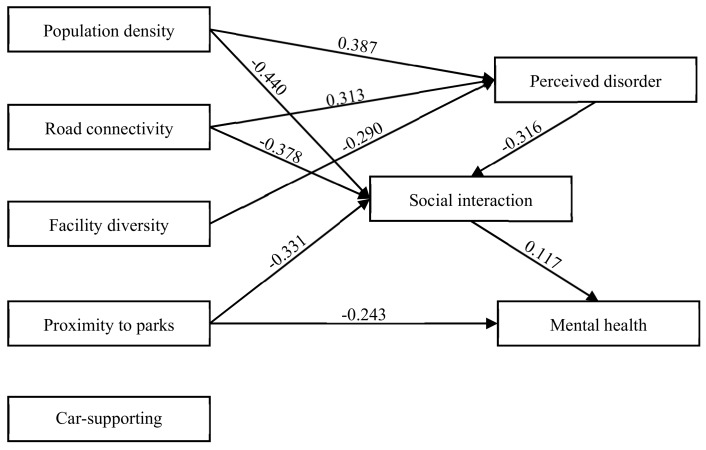
Relationships among built environment, perceived disorder, social interaction and mental health in Model 4. Note: Only standardized effects significant at the 0.1 level are presented. The results are adjusted for individual and household demographics, and neighborhood socioeconomic disadvantages.

**Table 1 ijerph-17-00013-t001:** Key sociodemographic attributes of the surveyed respondents (N = 1256).

Variable	Description	N/Mean	Pct. (%)/St. D.
Gender	Male	483	50.7
	Female	469	49.3
Age	18–35	316	25.2
	36–45	287	22.9
	46–55	232	18.5
	56–65	421	33.5
*Hukou*	Local residents	942	74.9
	Rural-to-urban migrants	315	25.1
Employment status	Full-time employment	728	58.0
	Others	528	42.0
Family monthly income	5000 RMB and below	242	19.3
	5001–15,000 RMB	725	57.7
	Above 15,000 RMB	289	23.0
Duration of residency (years)		13.79	11.73

Note: RMB = Renminbi, the official Chinese currency; 5000 and 15,000 RMB is equivalent to approximately 711 and 2134 US dollars, respectively.

**Table 2 ijerph-17-00013-t002:** Built environment characteristics and socioeconomic composition stratified by neighborhood types (N = 26).

Variable	Description	Commodity Housing Neighborhoods	Work-Unit Compounds	Policy Related Housing Neighborhoods	Mixed Neighborhoods
Population density	Average population in a 100 m × 100 m grid within an 800 m buffer	81.14	122.71	104.61	163.12
Road connectivity	Number of main road intersections (three-legged or greater) within an 800 m buffer	105.55	142.25	125.00	181.33
Facility diversity	A composite index calculated as the sum of the availability of the following facilities within an 800 m buffer: restaurants, malls, schools, banks, and post offices	3.3636	3.7500	3.4000	4.3333
Proximity to parks	Distance to the nearest park	0.85	0.77	1.08	0.84
Availability of rail stations	Whether there is a rail station within an 800 m buffer	27.3%	75.0%	40.0%	83.3%
Availability of parking lots	Whether there is a parking lot within an 800 m buffer	72.7%	25.0%	40.0%	66.7%
Proportion of rural-to-urban migrants	Number of rural-to-urban migrants/total population in the neighborhood	46.6%	32.3%	43.6%	33.5%
Proportion of elderly people	Number of elderly people 65 years old and above/total population in the neighborhood	8.1%	12.0%	7.4%	10.0%
Proportion of residents with high school degree and below	Number of people with high school degree and below/total population in the neighborhood	24.6%	37.4%	36.0%	28.5%

**Table 3 ijerph-17-00013-t003:** Unstandardized and standardized loadings for the measurement models.

Latent Constructs			
	Perceived Disorder	Social Interaction	Mental Health
Unsafe environment	1.000 (0.612)		
Noisy environment	1.438 (0.719)		
Lack of places/opportunities for interactions	1.591 (0.744)		
Neighbors visit		1.000 (0.867)	
Neighbors help		0.788 (0.664)	
Having meals with neighbors		0.842 (0.748)	
Exercising with neighbors		1.135 (0.875)	
Frequency of feeling anxious			1.000 (0.745)
Frequency of feeling stressed			1.105 (0.720)
Frequency of feeling tired			1.224 (0.844)
Frequency of having a headache			0.785 (0.685)

a. Coefficients with no standard errors have a fixed value of 1.0.

**Table 4 ijerph-17-00013-t004:** Multilevel structural equation models of mental health.

	Model 1	Model 2
	Mental Health	Mental Health
Neighborhood socioeconomic disadvantages		
Proportion of rural-to-urban migrants	−0.110 ** (0.042)	−0.120 ** (0.055)
Proportion of elderly people	−0.003 (0.037)	−0.021 (0.040)
Proportion of residents with high school degree and below	−0.058 ** (0.024)	−0.054 * (0.030)
Neighborhood built environment		
Population density		−0.229 (0.337)
Road connectivity		−0.330 * (0.200)
Facility diversity		0.254 (0.231)
Proximity to parks		−0.439 ** (0.207)
Car-supporting facilities		−0.273 (0.335)
Model fit		
χ^2^ (d.f.)	160.433 (35)	204.391 (50)
CFI	0.965	0.945
RMSEA	0.045	0.050
SRMR (individual level)	0.039	0.039
SRMR (neighborhood level)	0.037	0.044

Notes: All models are adjusted for gender, employment status, household monthly income, and duration of residency (results are not shown). Standardized coefficients with standard errors shown in brackets. * *p* < 0.10, ** *p* < 0.05, *** *p* < 0.01.

**Table 5 ijerph-17-00013-t005:** Multilevel structural equation models of perceived disorder, social interaction and mental health.

	Model 3		Model 4		
	Perceived Disorder	Mental Health	Perceived Disorder	Social Interaction	Mental Health
Perceived disorder		−0.110 (0.137)		−0.316 *** (0.047)	−0.103 (0.115)
Social interaction					0.117 ** (0.056)
Neighborhood socioeconomic disadvantages					
Proportion of rural-to-urban migrants	0.117 (0.103)	−0.109 * (0.059)	0.060 (0.090)	−0.220 ** (0.104)	−0.125 ** (0.050)
Proportion of elderly people	0.067 ** (0.031)	−0.015 (0.036)	0.085 ** (0.029)	−0.049 (0.071)	−0.005 (0.049)
Proportion of residents with high school degree and below	0.018 (0.013)	0.052 * (0.029)	0.020 (0.015)	−0.006 (0.005)	0.052 * (0.030)
Neighborhood built environment					
Population density	0.250 * (0.149)	−0.220 (0.344)	0.387 ** (0.160)	−0.440 ** (0.222)	−0.244 (0.347)
Road connectivity	0.267 * (0.157)	−0.318 * (0.206)	0.313 ** (0.130)	−0.378 * (0.210)	−0.287 (0.190)
Facility diversity	−0.226 * (0.158)	0.259 (0.237)	−0.290 * (0.187)	0.184 (0.183)	0.216 (0.220)
Proximity to parks	0.104 (0.156)	−0.440 ** (0.222)	0.157 (0.168)	−0.331 ** (0.146)	−0.243 * (0.151)
Car-supporting facilities	0.190 (0.194)	−0.245 (0.352)	−0.219 (0.177)	−0.014 (0.313)	−0.302 (0.304)
Model fit					
χ^2^ (d.f.)	391.278 (98)		645.334(163)		
CFI	0.919		0.914		
RMSEA	0.047		0.048		
SRMR (individual level)	0.036		0.045		
SRMR (neighborhood level)	0.046		0.049		

Notes: All models are adjusted for gender, employment status, household monthly income, and duration of residency (results are not shown) Standardized coefficients with standard errors shown in brackets. * *p* < 0.10, ** *p* < 0.05, *** *p* < 0.01.

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
