# Peer review of "The Anatomy of Health-Supportive Neighborhoods: A Multilevel Analysis of Built Environment, Perceived Disorder, Social Interaction and Mental Health in Beijing"

_ijerph, 2019, doi:10.3390/ijerph17010013_

Round 1

Reviewer 1 Report

This paper used a health survey conducted from Beijing, China to understand the relationships among built environment, perceived disorder, social interaction, and mental health. Overall, this research is interesting. Research questions and objectives are well defined. Research methodology is valid. The following comments must be addressed before acceptance.

a detailed figure of how SEM models were constructed should be shown for a better clarification of the model structure. another concern is for the data validity. is the sample representative for Beijing, which has more than 21 million people. and was the sample representative for all the neighborhoods in Beijing?  how could results be applied to other cities in China? Beijing is a megacity which has many unique urban features and is Beijing representative to other cities in China? The policy conclusion suggests that pedestrian oriented environment should be promoted-- however, there is no variable in the study that reflects the pedestrian oriented environment. 

Reviewer 2 Report

This study aims to clarify the complex relationships between the built environment, its perception (tranquility, security, residential areas, ...), social interaction (social relations between neighbors) and mental health (anxiety, fatigue, headache , ...) in socioeconomically disadvantaged neighborhoods of a developing megacity such as Beijing. The objective of the article is successfully achieved after a detailed reading of it.

From the methodological point of view, the study is impeccable. The use of 1300 health surveys conducted (of which 1256 were valid) in Beijing in 2017 and the use of multilevel structural equation models is totally adequate and relevant. Perhaps it would be interesting to present a digital link to the surveys in order to identify them.

In this regard, perhaps, to facilitate statistical comparison with other countries, the dollar in the "Family monthly income" section should be used as the official currency in Table 1. On the other hand, it would be interesting to know the origin of the population density, road connectivity, proximity to parks and availability of railway stations and parking lots.

Some terms may require some additional explanation to facilitate comparison with other urban realities, such as "Work-unit compounds", "communities" or "point-of-interest".

On the other hand, lines 323, 324 and 325 state the following "Neighborhood socioeconomic disadvantages were measured by the proportion of rural-to-urban migrants and old people aged over 65 years old in the neighborhood. These two indices have been used in many Chinese urban studies to depict the extent of disadvantaged social contexts. " In relation to this there are numerous contributions that suggest how urban disadvantage would be linked to more socio-economic and demographic indicators than those used in this research. Similarly, the results of the research could be linked to what in the scientific literature has been raised as "space justice." We recommend the use of some contributions in this regard such as those of Moreno et al. (2016), Geoforum, No. 69, Palacios et al. (2018), Urban Science, 2 (4), or Vazquez et al. (2018) in Bargelli, E. and Heitkamp, ​​T. (Eds.): New developments in southern european housing.

The conclusions are especially relevant, having carried out a reflexive and purposeful analysis of reality. In fact, proposals are proposed concerning urban planners and administrators on how to build neighborhoods that support the health and development of healthy cities.
